# Organisational and extraorganisational determinants of volume of service delivery by English community pharmacies: a cross-sectional survey and secondary data analysis

Mark Hann,[1] Ellen I Schafheutle,[2] Fay Bradley,[2] Rebecca Elvey,[2,3] Andrew Wagner,[4] Devina Halsall,[2,5] Karen Hassell,[2,6] Sally Jacobs[2]

For numbered affiliations see end of article.

**Correspondence to**
Dr Sally Jacobs;
sally.jacobs@manchester.ac.uk

## ABSTRACT

**Objectives** This study aimed to identify the organisational and extraorganisational factors associated with existing variation in the volume of services delivered by community pharmacies.

**Design and setting** Linear and ordered logistic regression of linked national data from secondary sources—community pharmacy activity, socioeconomic and health need datasets—and primary data from a questionnaire survey of community pharmacies in nine diverse geographical areas in England.

**Outcome measures** Annual dispensing volume; annual volume of medicines use reviews (MURs).

**Results** National dataset (n=10 454 pharmacies): greater dispensing volume was significantly associated with pharmacy ownership type (large chains>independents>supermarkets), greater deprivation, higher local prevalence of cardiovascular disease and depression, older people (aged >75 years) and infants (aged 0–4 years) but lower prevalence of mental health conditions. Greater volume of MURs was significantly associated with pharmacy ownership type (large chains/supermarkets>>independents), greater dispensing volume, and lower disease prevalence. Survey dataset (n=285 pharmacies; response=34.6%): greater dispensing volume was significantly associated with staffing, skill-mix, organisational culture, years open and greater deprivation. Greater MUR volume was significantly associated with pharmacy ownership type (large chains/supermarkets>>independents), greater dispensing volume, weekly opening hours and lower asthma prevalence.

**Conclusions** Organisational and extraorganisational factors were found to impact differently on dispensing volume and MUR activity, the latter being driven more by corporate ownership than population need. While levels of staffing and skill-mix were associated with dispensing volume, they did not influence MUR activity. Despite recent changes to the contractual framework, the existing fee-for-service reimbursement may therefore not be the most appropriate for the delivery of cognitive (rather than supply) services, still appearing to incentivise quantity over the quality (in terms of appropriate targeting) of services delivered. Future research should focus

## Strengths and limitations of this study

- ► This paper is the first to examine systematically a range of organisational features associated with the volume of services provided by community pharmacies.
- ► The study took the novel approach of analysing linked national secondary datasets (pharmacy activity, socioeconomic and health need), alongside organisational survey data for a subsample of pharmacies.
- ► Triangulation of methods strengthens the validity of the findings that will help service commissioners and policymakers understand the organisational context influencing service delivery by private sector organisations in a mixed healthcare economy.
- ► One limitation of this study relates to non-random sampling of pharmacies and the non-participation of four large pharmacy chains, both of which threaten generalisability.
- ► The second limitation arises from a low survey response rate and the possibility of non-response bias.

on the development of quality measures that could be incorporated into community pharmacy reimbursement mechanisms.

## INTRODUCTION

Over the past 30 years, many developed countries that have historically favoured public sector funding and provision of healthcare have sought to develop a mixed economy of healthcare provision, involving private and third-sector organisations in developing infrastructure and delivering services.[1] The National Health Service (NHS) in the UK is a publicly funded institution providing comprehensive healthcare that is mostly free at the point of delivery. Since the 1980s,

successive UK governments have introduced a raft of promarket policies that have sought to increase the range and diversity of healthcare provider organisations to increase patient choice, access and service quality while saving money for the public purse.[2] However, concerns have been raised over the quality and safety of patient care[3] and the motivations of managers[4] in private sector organisations where there is a need to balance delivery of healthcare with generation of profit.

Similar to other countries, community pharmacies in the UK are for-profit organisations delivering services under contract to the NHS alongside a range of products and services available for customer purchase, from medicines and professional healthcare to cosmetics and groceries.[5] Community pharmacies take a number of organisational forms under different types of ownership, from large national chains and supermarkets, small and medium-sized chains through to independently owned pharmacies. They are established private sector providers of public services and provide an ideal exemplar in which to examine organisational influences on the delivery of healthcare in this sector.

International trends in pharmacy practice have seen a shift from preparation and supply of medicines towards a more clinical role encompassing medicine-related and healthcare advice, supported self-care and public health service delivery.[6] In recognition of this, the contractual framework for NHS community pharmacy services in England has, since 2005, recognised three levels of service: essential services (dispensing and repeat dispensing) provided by all community pharmacies, advanced services (medicines use reviews (MURs), the new medicines service (introduced in 2011) and influenza vaccinations (introduced in 2015)) and locally commissioned services (minor ailments schemes, public health services and support for self-management of long-term conditions). To encourage pharmacies to provide a wider range of services, when the 2005 contractual framework was introduced, the remuneration for dispensing was top sliced, and a proportion of the available budget was allocated to the remuneration of advanced services, with individual pharmacies permitted to claim for up to 400 MURs per annum (250 during 2005/2006) on a fee-for-service basis.

MURs offer patients the opportunity to discuss their knowledge, understanding and use of medications. They aim to help patients manage their medicines more effectively, identifying any side effects or interactions, improving adherence, reducing medicines waste and providing feedback to prescribers. While pharmacists generally embrace the principles behind MURs, viewing them as an opportunity to use their clinical skills and knowledge and to offer benefit to patients,[7 8] their implementation and operation have been problematic. Public awareness of the service remains low,[8] and evidence produced soon after their introduction suggests that they are not valued by general practitioners (GPs; family physicians)[9] who have little engagement with the process[10] limiting their potential impact. Early research also demonstrated that the

volume of MUR provision by pharmacy chains was greater than that of independently owned pharmacies,[11 12] with larger organisations pressurising employee pharmacists to meet stringent targets, prioritising service quantity over quality[12] and compromising the professional judgement of pharmacists.[13] The research by Bradley et al[12] also suggested that the volume of MURs conducted was inversely related to local health need raising questions about the extent to which they have been appropriately targeted.

Other organisational factors have been shown to be associated with the uptake and delivery of extended community pharmacy services worldwide. These include work overload and conflicting workloads, staffing and skill-mix, organisational culture and leadership, relationships with local GPs, the physical environment (particularly consultation space and privacy), equipment and technology.[12 14–21] The role of organisational culture within community pharmacy, particularly the inherent dichotomy between business and professional values, varies in relation to other organisational characteristics (eg, ownership type, with larger corporate pharmacy chains tending to have a greater business orientation than independent pharmacies that tend to be more patient focused)[15] and may be central to the difficulties experienced by pharmacies in implementing services such as MURs.

In an attempt to counteract concerns that MURs had not always been offered to those most likely to benefit, a 2011 revision to the contractual framework required that at least 50% of all MURs were targeted towards patients in three groups: those taking a defined list of high-risk medicines, following hospital discharge and those with respiratory disease. This was extended in 2015 to include a fourth target group, cardiovascular patients taking four or more medicines, and the required proportion of targeted MURs increased to 70%.

Based on findings from a larger study of variation in clinical productivity in English community pharmacies,[22] this paper aims, first, to update the analysis conducted not long after the introduction of MURs by Bradley et al[12] to examine whether pharmacy ownership remains a key predictor of service volume and if there has been any change in the association with population need following the introduction of targeted MURs in 2011. Second, this paper seeks to extend that analysis, looking past ownership type at other organisational and extraorganisational factors associated with service volume.

## METHODS

Two sets of regression analyses were conducted. For a broad picture of dispensing and MUR activity across England, routinely collated community pharmacy activity data, socioeconomic and health need data were obtained from national datasets. To obtain more detailed information about individual pharmacies, a questionnaire survey was conducted in nine diverse primary care administrative

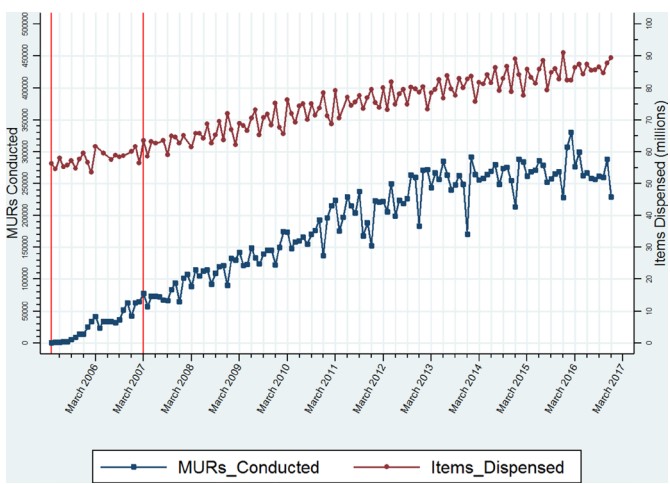

**Figure 1** Monthly dispensing and MUR activity in community pharmacies in England (April 2005–December 2016). MURs, medicines use reviews.

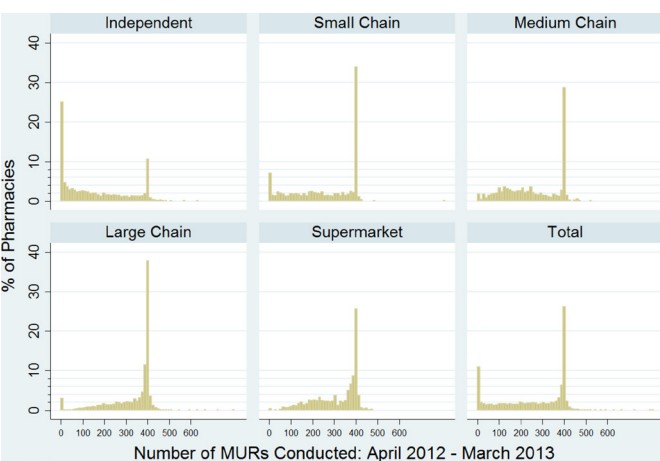

**Figure 2** Number of MURs conducted in community pharmacies in England by pharmacy ownership type (April 2012–March 2013). MURs, medicines use reviews.

areas. All inferential analyses were conducted using STATA software (V.13).[23]

Ethical approval was obtained from the National Research Ethics Service Committee West Midlands – Edgbaston (13/WM/0137), endorsed by the University of Manchester Research Ethics Committee (13025).

### National data sources and analyses

Aggregated monthly data (already in the public domain) on numbers of items dispensed and numbers of MURs conducted across all pharmacies in England were downloaded directly from the Pharmaceutical Services Negotiating Committee website[24] and from their data archive,[25] up to the end of 2016. Dispensing data were not available for the full 2012 calendar year so these data were downloaded directly from the NHS Business Services Authority (BSA) website.[26] Using this information collectively, an extension of figure 1 in Bradley *et al*[12] was produced to illustrate items dispensed and MURs conducted for March 2005–December 2016 (figure 1).

In addition, and with appropriate approval, we obtained the same data from the NHS BSA, by pharmacy, for the period April 2011–October 2013. These data were transferred using a secure portal and linked, by pharmacy premises postcode and unique NHS organisational code, to determinants of the demographic, socioeconomic and health needs status of the population within the immediate individual pharmacy locality (super output area) obtained from national secondary datasets. These included: (A) the income deprivation domain of the 2010 English Indices of Multiple Deprivation (IMD)[27]; (B) Office for National Statistics 2011 Census data,[28] including the proportion of the local population with a self-reported limiting long-term illness and within certain age bands (eg, 0–4 years; ≥75 years); and (C) 2011/2012 Quality and Outcomes Framework disease prevalence data for conditions for which community pharmacies (can) provide clinical services (eg, coronary heart disease (CHD)/mental health (MH) condition (schizophrenia,

bipolar affective disorder and other psychoses)/depression/asthma).[29] This combined dataset also contained information on characteristics of the pharmacy, such as pharmacy type: a categorisation of pharmacy ownership was created from NHS head office codes as either independent (<6 branches), small chain (6–25 branches), medium chain (26–200 branches), large chain (>200 branches) or supermarket consistent with that used in other analyses. The BSA data on MURs was used, for the financial year April 2012–March 2013, to produce an updated version of figure 2 in Bradley *et al*,[12] which depicts numbers conducted annually by pharmacy type (figure 2, n=11 033). Then, using either linear (dispensing) or ordered logistic (MURs) regression, we determined which of the characteristics mentioned above had the strongest association with volume of service delivery at a national level (table 1; n=10 454, following exclusion of pharmacies with extremely high or low annual dispensing volumes relative to other pharmacies (outliers, n=60) and those for which a full set of (linked) data for 2012/2013 were unavailable).

All data were stored securely on an encrypted university network drive only accessible to the statistician conducting the analysis (MH).

### Survey of community pharmacies and analyses

The variables in the combined dataset above broadly describe the geography/demography of the area in which the pharmacy is located but do not—with the exception of pharmacy ownership type—represent characteristics of the pharmacy. To ascertain more detailed information about the organisational characteristics of individual pharmacies, we conducted a survey of community pharmacies in nine primary care administrative areas across England, purposively selected to cover a geographically diverse range of affluent/deprived areas of dense/sparse populations. With permission from local pharmaceutical committees, contact details for all pharmacy premises, including postcode and unique organisational identifier,

**Table 1** Parameter estimates from regression models of national data

| | (Log of) Dispensing volume | | MURs conducted | | |
|---|---|---|---|---|---|
| | Standardised coefficient | p Value | OR | 95% CI | p Value |
| Pharmacy type | | | | | |
| Independent | baseline | <0.001 | baseline | (3.891 to 5.152) | <0.001 |
| Small chain | 0.019 | | 4.477 | (4.271 to 5.611) | |
| Medium chain | −0.023 | | 4.895 | (10.568 to 12.889) | |
| Large chain | 0.126 | | 11.671 | (10.123 to 13.229) | |
| Supermarket | −0.158 | | 11.573 | | |
| IMD score* | 0.079 | <0.001 | 0.998 | (0.996 to 1.001) | 0.173 |
| % Aged 0–4* | 0.037 | 0.003 | 1.005 | (0.982 to 1.028) | 0.672 |
| % Aged ≥75 years* | 0.092 | <0.001 | 1.004 | (0.994 to 1.015) | 0.417 |
| % with CHD† | 0.108 | <0.001 | 0.762 | (0.711 to 0.818) | <0.001 |
| % with MH condition† | −0.113 | <0.001 | 0.801 | (0.625 to 1.026) | 0.079 |
| % with Depression† | 0.034 | 0.005 | 0.938 | (0.901 to 0.977) | 0.002 |
| % with Asthma† | 0.008 | 0.640 | 0.873 | (0.801 to 0.953) | 0.002 |
| log (Dispensing vol.) | - | - | 11.053 | (9.438 to 12.944) | <0.001 |

*†These 'groups' of variables were moderately intercorrelated. Variance inflation factors were computed by STATA, which indicated that no variable(s) needed to be removed from the model due to collinearity.
CHD, coronary heart disease; IMD, Indices of Multiple Deprivation; MH, mental health; MUR, medicines use review.

were obtained by the research team from healthcare commissioners in each area. The head offices of nine of the largest pharmacy chains, who together make up approximately 50% of the community pharmacy market, had been approached individually to obtain permission to survey their pharmacies. Four had declined to participate in the survey, and these pharmacies were excluded from the 'sampling frame', which otherwise included all pharmacies providing services in these areas (n=817). This sample had 90% (77%) power to detect a correlation as small as 0.16 between organisational factors and service volume based on a 5% level of statistical significance and assuming a non-response rate of 50% (66%).

An eight-sided self-completion questionnaire and reply-paid envelope was distributed by post (with an option to complete online) to all 817 pharmacies (addressed to the pharmacy manager) in February 2014. Two postal reminders (including additional copies of the questionnaire) were sent at 3-week intervals. The questionnaire collected information on: pharmacy ownership type, location, opening hours, staffing, skill-mix, working patterns and organisational culture. Details of the characteristics collected by this survey can be found in table 2. Most items were developed and validated by the research team through previous surveys, from existing evidence of organisational characteristics that have been shown to influence care provision.[11–13 15 16 18–21 30] Organisational culture was measured using the Pharmacy Service Orientation (PSO) tool[31] scored on the basis of the mean of three 1–10 semantic differential scales whereby respondents rated their pharmacy's orientation (patient vs medicine), focus (quality vs quantity) and pharmacists work

(professional vs technical). The questionnaire was piloted with nine community pharmacy managers, recruited through existing contacts, using cognitive interviewing techniques.[32]

The survey data were linked by pharmacy premises postcode and unique NHS organisational code to service activity data for the period April 2012–March 2013 (the most recent financial year for which a full set of data was available) obtained from the NHS BSA and determinants of the demographic, socioeconomic and health needs status of the population obtained from the national datasets described above.

A series of univariable linear (for dispensing volume) or ordered logistic (for MURs conducted) regression models were fitted to determine which pharmacy-level organisational variables and/or areal-specific demographic, socioeconomic and health needs variables were associated with each outcome. A conservative p value of 0.2 was employed to indicate a significant association. Independent variables meeting this criterion were then included in an appropriate multivariable regression model to determine if their association persisted on controlling for other factors. Primary care administrative area (treated as a fixed effect) and pharmacy ownership type were added to the model at this point. Variables were retained in the 'final' model, along with ownership type and study area and after removal of collinear ones, if significance at p=0.05 was achieved. Probability weights were calculated as the ratio of the number of each pharmacy type within the locality to the number of each pharmacy type within the locality who responded to the survey (ie, the inverse of the probability of response) and were applied,

**Table 2** Organisational characteristics of community pharmacies responding to the survey

| Variable | Category* or summary statistic† | Completed responses | |
| --- | --- | --- | --- |
| | | N (%) or statistic | Total |
| Job title (respondent) | Owner | 55 (20.2) | 273 |
| | Manager | 168 (61.5) | |
| | Other pharmacist | 50 (18.3) | |
| Type of pharmacy | Independent (<6 stores) | 111 (40.1) | 277 |
| | Small chain (6–25 stores) | 41 (14.8) | |
| | Medium chain (26–200 stores) | 9 (3.3) | |
| | Large chain (>200 stores) | 91 (32.9) | |
| | Supermarket | 25 (9.0) | |
| Geographical location | City centre | 18 (6.6) | 274 |
| | Large town | 43 (15.7) | |
| | Small town | 81 (29.6) | |
| | Suburb | 96 (35.0) | |
| | Village/rural | 36 (13.1) | |
| Pharmacy open ≥3 years | No | 22 (8.0) | 274 |
| | Yes | 252 (92.0) | |
| Healthy living pharmacy‡ | No | 192 (72.2) | 266 |
| | Yes | 74 (27.8) | |
| Pharmacy contract held§ | Standard 40 hours | 230 (84.2) | 273 |
| | 100 hours | 30 (11.0) | |
| | Other | 13 (4.8) | |
| Opening hours/week | Mean (SD) | 55.8 (17.4) | 274 |
| | Median (IQR) | 50 (45–56) | |
| | Range | 36–104 | |
| No. staff working on typical day | Mean (SD) | 5.4 (2.2) | 268 |
| | Median (IQR) | 5 (4–6.5) | |
| | Range | 1–18 | |
| Pharmacists working on a typical day | 1 | 219 (80.5) | 272 |
| | ≥2 | 53 (19.5) | |
| Registered pharmacy technician (typical day) | No | 159 (58.2) | 273 |
| | Yes | 114 (41.8) | |
| Accuracy checking technician (typical day) | No | 209 (75.7) | 276 |
| | Yes | 67 (24.3) | |
| Use of locums¶ | Not regularly | 118 (42.6) | 277 |
| | Regularly | 159 (57.4) | |
| Pharmacy manager is a pharmacist? | No | 30 (10.9) | 274 |
| | Yes | 244 (89.1) | |
| Work pattern of main pharmacist | Standard hours (08:00–18:00) | 165 (59.6) | 277 |
| | Non-standard | 112 (40.4) | |
| Average daily working hours of main pharmacist | Mean (SD) | 9.2 (1.1) | 270 |
| | Median (IQR) | 9.0 (8.5–9.75) | |
| | Range | 5–15 | |
| Organisational culture (PSO)** | Mean (SD) | 7.3 (1.4) | 269 |
| | Median (IQR) | 7.5 (6.5–8.2) | |
| | Range | 3.5–10 | |

Continued

**Table 2** Continued

| Variable | Category* or summary statistic† | Completed responses | |
|---|---|---|---|
| | | N (%) or statistic | Total |
| Relationship with nearest GP surgery | Very good | 136 (49.1) | 277 |
| | Good | 83 (30.0) | |
| | Satisfactory/poor/none | 43 (15.5) | |
| | No GP surgery identified | 15 (5.4) | |

*Categorical variables (eg, job title; type of pharmacy) are summarised by the number and percentage of responses in each category.

†Continuous variables (eg, opening hours per week; number of staff on a typical day) are summarised using their mean (SD), median (IQR) and range.

‡Healthy living pharmacies have been accredited for consistently demonstrating a healthy living ethos and a proactive approach to health and health improvement.

§A UK pharmacy is normally contracted to be open for 40 core hours (or 100 for those that have opened under the former exemption from the control of entry test). Other contract types include 'essential small pharmacies' and pharmacies contracted under other 'local pharmaceutical services' agreements.

¶*Locum tenens* or substitute pharmacists.

**Pharmacy Service Orientation (PSO) tool[30] scored on the basis of the mean of three 1–10 semantic differential scales whereby respondents rate their pharmacy's orientation (patient vs medicine), focus (quality vs quantity) and pharmacists work (professional vs technical).

GP, general practitioner.

in order to make the sample of respondent pharmacies more representative of the population of pharmacies in their area.

## RESULTS
### Trends in monthly dispensing volume and MURs conducted

The number of MURs conducted has continued to rise steadily year-on-year from their inception in April 2005 (figure 1). By the end of the data period reported in Bradley *et al* (March 2007—indicated by the vertical red lines), around 80 000 MURs per month were being conducted nationally. By March 2010, this number had doubled, and by March 2013, it had trebled. However, thereafter, there are signs that the rate of increase has started to decline, despite the peak in excess of 300 000 MURs conducted in the early months of 2016. There are noticeable 'seasonal effects', with dramatic declines in December followed by sharp increases in January and February as the financial year draws to a close at the end of March.

The volume of items dispensed on a monthly basis also increased steadily year-on-year but not at the same rate as for MURs. The peak was in December 2015, when in excess of 90 million items were dispensed .

### MURs conducted by pharmacy ownership type: April 2012–March 2013

Figure 2 shows the distribution of the number of MURs conducted annually by individual pharmacies, both overall (total) and by pharmacy type. Of the 11 033 pharmacies for which we have data, 3070 (27.8%) conducted between 365 and 400 MURs annually, while a further 1294 (11.7%) conducted in excess of 400 MURs (despite no remuneration being available for MURs conducted beyond the 400th). Conversely, 911 pharmacies (8.3%) reported conducting zero MURs, with a further 385

(3.5%) conducting between 1 and 12 (ie, an average of up to one per month).

Independent pharmacies had a different pattern of MUR conduct to chains (of any size) and supermarkets: more than one-quarter conducted fewer than 12 MURs annually, while the 'spike' at around 400 was much less pronounced than for other pharmacy types (and overall).

### Associations with service volume in national data

As is evident from figure 2, the distribution of the total number of MURs conducted per year was fairly uniform but with a very pronounced spike at 400 and another, smaller, spike at zero. The median value was 316 with an IQR of 146–397. Given this unusual distribution, there was a risk that using, for example, quintiles as a means of categorisation would have produced some groupings closely clustered around 400. This outcome variable was therefore categorised in a more meaningful way as follows: 0–12, 13–200, 201–365 and >365 (equivalent to <1 per month up to >1 per day) and ordered logistic regression used in its analysis.

The distribution of annual dispensing volume was positively skewed (median=72 325 items per year; IQR=49 569–103 213; skewness=1.5; kurtosis=7.0), with 10% of pharmacies dispensing in excess of 140 000 items per year. Accordingly, linear regression of the logarithmic value of dispensing volume was used in its analysis to eliminate this long right-hand tail. With such large numbers of pharmacies in the dataset (n=10 454), even very small associations are likely to be significant, and p values will not be particularly informative. In addition, direct interpretation on the logarithmic scale is not straightforward and, therefore, parameter estimates were standardised so that their magnitude is directly comparable.

Pharmacy ownership type had a strong overall association with dispensing volume in the national dataset ($F_{(4,}$

$_{10\,442)}$=158.5; p<0.001) (table 1). Individually, large pharmacy chains and supermarkets had the largest absolute standardised coefficients, but of opposite sign, suggesting that the former dispensed a greater volume of items, whereas the latter dispensed fewer items (compared with independent pharmacies).

Of the area-level variables, the strongest associations with dispensing volume were that a higher local prevalence of CHD was associated with a *greater* dispensing volume at the pharmacy, whereas a higher prevalence of MH conditions was associated with a *lesser* dispensing volume. Dispensing volume was also greater in pharmacies whose local population was ageing (greater % aged≥75) and, independently, more deprived.

Very strong associations existed between the number of MURs conducted and both pharmacy ownership type ($\chi^2_{(4)}$=26580.7; p<0.001) and dispensing volume (Z=29.8; p<0.001). Compared with independent pharmacies, all other organisational types conducted significantly more MURs over the course of the financial year, with large chains and supermarkets the most active in this respect. A greater dispensing volume was also associated with a greater annual number of MURs conducted.

The strength of association with other variables was generally much weaker, although they were measured on very different scales and at a different level of geography (they relate to the area in which the pharmacy is located, rather than to the pharmacy itself). ORs for the prevalence of long-term conditions in the local primary care administrative area were all less than unity, indicating that increased disease prevalence among the local population—but not necessarily the population 'attending' a particular pharmacy (the ecological fallacy)—was associated with the conduct of fewer MURs. The level of local deprivation and age structure were not significantly associated with the number of MURs conducted, even with n≈10 500 pharmacies.

## Associations with service volume in the survey data

Of the 817 questionnaires distributed, 285 were returned completed: 260 by post and 25 online. A further nine were returned undelivered. Eight questionnaires completed by distance selling pharmacies (those only dispensing by post and not face to face) were removed. The total valid response rate was 277/800 (34.6%). The applied probability weights varied from 1.3 to 6.5, which is a narrow range.

Descriptive statistics for each of the key independent variables used in the analysis of community pharmacy activity in the survey dataset are presented in table 2. The proportion of independently owned pharmacies in this sample of respondents was comparable with the national figure of 38.9% in March 2014.[33] As a measure of organisational culture, the mean (SD) PSO score in this sample was 7.3 (1.4).

The large list of independent variables available from the survey (>40) would be problematic for simultaneous model estimation in this size of sample and, therefore, we only considered, in the multivariable analysis, variables for which p<0.2 in a univariable analysis. Variables that were non-significant (ie, p≥0.05) or were clearly collinear in the multivariable model were subsequently removed in order to achieve the most parsimonious model. Here, for analytical purposes, due to the smaller numbers (compared with the national dataset), small- and medium-sized pharmacy chains were combined, as were large chains and supermarkets.

To maintain consistency, the outcome variable measuring MUR conduct was analysed using the categorisation created for the national data (0–12, 13–200, 201–365 and >365): ordered logistic regression was used. The distribution of annual dispensing volume in the survey data was positively skewed (median=74 187 items per year; IQR=53 869–104 810; skewness=1.6; kurtosis=7.7) and linear regression of the logarithmic value of dispensing volume was used.

The results of the multivariable analyses (final models only) are reported in table 3; it is evident that variables associated with either dispensing volume or MURs conducted were mutually exclusive.

Staffing levels and skill-mix had important associations with dispensing volume: pharmacies with higher dispensing volumes were significantly more likely also to employ two or more pharmacists, have a registered pharmacy technician and, independently, an accuracy checker. Increasing average daily working hours of the main pharmacist was also associated with a greater dispensing volume. Organisational culture proved to be significantly associated with dispensing volume: pharmacies with higher dispensing volumes were perceived as having a greater focus on quantity, technical work and the medicine than on quality, professional work and the patient. Pharmacies that had been open for 3 or more years also had significantly higher dispensing volumes. The level of deprivation in the local population was also significantly associated with dispensing volume, but health need variables were not retained in the final model due to collinearity with the IMD score their intercorrelation.

The volume of MURs conducted by these pharmacies was strongly positively associated with annual dispensing volume. Controlling for this in the multivariable regression, pharmacy ownership type had the strongest association of the organisational variables with volume of MUR provision, with chains, particularly large chains and supermarkets, conducting significantly more MURs annually than independent pharmacies. Although a number of organisational factors were univariably associated with volume of MURs, the only other organisational factor remaining in the final multivariable model was weekly opening hours: pharmacies with longer weekly opening hours conducted more MURs. The volume of MURs conducted was not related to the level of deprivation in the local population (IMD score), although a significant negative association was seen with the prevalence of asthma.

**Table 3** Significant associations from regression models of the community pharmacy survey data

| | Dispensing volume | | | MURs conducted | | |
|---|---|---|---|---|---|---|
| | Coefficient | 95% CI | p Value | OR | 95% CI | p Value |
| Pharmacy type | | | | | | |
| Independent | | | | Baseline | | <0.001 |
| Small chain/medium chain | | | | 2.67 | 1.39 to 5.15 | |
| Large chain/supermarket | | | | 4.86 | 2.63 to 8.96 | |
| Pharmacy open ≥3 years | 0.3147 | 0.1233 to 0.5062 | 0.001 | | | |
| Weekly opening hours | | | | 1.25 | 1.06 to 1.49 | 0.010 |
| >1 Pharmacist (on a typical day) | 0.0949 | 0.0188 to 0.1710 | 0.015 | | | |
| Registered pharmacy technician (typical day) | 0.1120 | 0.0599 to 0.1641 | <0.001 | | | |
| Accuracy checking technician (typical day) | 0.1655 | 0.1112 to 0.2197 | <0.001 | | | |
| Average daily working hours of main pharmacist | 0.0314 | 0.0086 to 0.0541 | 0.007 | | | |
| Organisational culture (PSO) | −0.0194 | −0.0373 to −0.0014 | 0.035 | | | |
| IMD score | 0.0164 | 0.0024 to 0.0304 | 0.022 | | | |
| % with Asthma | | | | 0.42 | 0.21 to 0.86 | 0.018 |
| log(dispensing volume) | | | | 5.88 | 1.76 to 19.6 | 0.004 |

IMD, Indices of Multiple Deprivation; MUR, medicines use review; PSO, Pharmacy Service Orientation.

## DISCUSSION

In seeking to identify the organisational and extraorganisational determinants of service volume (dispensing and MURs) in English community pharmacies, this paper has updated and extended the analysis conducted by Bradley *et al* soon after the introduction of MURs in 2005; since then, contractual changes have sought to better target their provision. Looking past ownership type as the sole defining organisational feature of community pharmacies, this paper has, for the first time, examined a range of organisational characteristics, including organisational culture, staffing and skill-mix and working patterns. By analysing both national activity data and a subset of data from a pharmacy survey, this study has demonstrated that dispensing volume is driven by local population need and is associated with a quantity/technical work/medicine-focused organisational culture and the employment of a larger number and greater range of pharmacy staff. Whereas volume of MURs conducted, while driven by dispensing volume, is still independent of (or inversely related to) local need and is strongly dependent on pharmacy ownership type over and above any other organisational variable measured.

A particular strength of this study is the examination of national pharmacy activity data alongside detailed organisational characteristics survey data for a subsample of community pharmacies in England. Adopting this dual approach has identified those findings consistent between the survey dataset and national dataset analyses while exploring a greater number of explanatory variables than would be possible otherwise. The study does, however, have a number of limitations. First, the survey sample of community pharmacies was not selected randomly. Study sites were originally selected purposively

to capture geographic and socioeconomic diversity and pragmatically on the basis of agreed access to pharmacy data and contact lists. While it is not possible to state that the findings are statistically generalisable beyond the nine study sites, the distribution of pharmacy types and activity levels are comparable with national figures. Second, non-participation in the survey by four of the nine largest community pharmacy chains for reasons of commercial sensitivity, pharmacists' workload and the impact of the concurrent pharmacy commissioning reorganisation constituted a further threat to generalisability. Despite the overall proportion of large chain pharmacies in the survey dataset mirroring national figures, variation in the other organisational characteristics in the sample may be limited. A third limitation arises from the low response rate to the survey and the resultant possibility of non-response bias. Following discussions with the larger chains, this was not unexpected for the reasons stated above, and great efforts were therefore made to maximise response rates through seeking study endorsement by stakeholder organisations including trade bodies and local pharmaceutical committees and two postal follow-ups to non-responders. However, although low response rate raised the risk of non-response bias, power was maintained: even at a non-response rate of 66%, the study was powered at 77% to detect a correlation as low as 0.16 between organisational factors and service volume. Finally, the method used to categorise the MUR outcome variable, while more meaningful than the narrower groupings produced by using for example, quintiles, may have resulted in different significant findings.

As one would expect, the findings have demonstrated that local population need, in terms of deprivation, age (proportion of older people/infants) and prevalence

of some long-term conditions (CHD and depression), is significantly and positively associated with dispensing volume. However, despite contractual requirements being introduced in 2011 to target 50% of MURs at high need patient groups (including those with respiratory disease) in the year of analysis (2012/2013), there remained no such association between local population need and MUR activity. Indeed, the findings show a significant negative association with the prevalence of a number of long-term conditions, including asthma, in the national and survey datasets. In this respect, there has been no change since the Bradley *et al* analysis of MURs conducted in 2006/2007,[12] their second year of operation, which demonstrated an inverse relationship between volume of MURs and levels of deprivation (IMD) and the proportion of those with a limiting long-term illness within the local population. This inverse relationship is counterintuitive, particularly given the positive association with dispensing volume that is associated with need (discussed below). However, the mechanism is far from clear. One factor may be the 400/year remuneration cap on MURs that stops pharmacies in areas with greater health needs providing a greater volume of services to meet that need. The problem of insufficient targeting of pharmacy-led medicine review services was similarly raised in investigations of the Australian Home Medicines Review programme[34] and the Canadian MedsCheck programme.[35] Assuming that the appropriate targeting of MURs is one dimension of the quality of such services, this calls into question the extent to which the quantity of services delivered is prioritised over their quality.

Also reflecting the earlier analysis by Bradley *et al*,[12] pharmacy ownership type (eg, independent and chain) was the strongest predictor of service volume. Analysis of the national dataset suggests that large chains dispense the highest volume of prescription items and supermarkets the least, yet both pharmacy types deliver significantly elevated volumes of MURs compared with independent pharmacies. The volume of MURs conducted was strongly associated with dispensing volumes overall, suggesting that in high dispensing volume pharmacies with a higher patient footfall (and more staff and more diverse skill-mix), there are increased opportunities to offer MURs. However, comparably high numbers of MURs conducted in the low dispensing volume supermarket pharmacies suggests other influences may be at play. Previous qualitative research has suggested that pressures on pharmacists to meet targets for MURs are more stringent in these larger organisations,[12 13] and this may be a contributing factor.

The above might suggest organisational culture as a possible determinant of service volume. Using a tool that captures the perceived balance in a pharmacy's orientation between patient or medicine, quality or quantity and professional or technical work,[31] the organisational culture of high dispensing pharmacies was demonstrated to be more closely aligned to medicine, quantity and technical work than low dispensing pharmacies. However, no

significant association was demonstrated between organisational culture and volume of MURs conducted. It is possible that the high proportion of large chain pharmacies reaching the 400/year cap on MURs is masking other associations, with larger organisations enforcing this cap as a target for all pharmacies, irrespective of other factors. It may be that the absence of four national chains from the sample has limited the variation in organisational culture seen in this category of pharmacies. Indeed, the mean PSO value was higher (ie, more closely aligned to the patient, quality and professional work than to the medicine, quantity and technical work) than the mean value of 6.3 (1.8) found in a previous survey of 903 English community pharmacists (personal communication, Jacobs, University of Manchester, 2012) suggesting that pharmacies in non-participating chains may have a relatively low PSO score. However, without a more in-depth exploration of the organisational culture extant in these different types of pharmacy, it is difficult to determine the underlying mechanisms. In the UK, the nine largest chains (>200 branches) differ in terms of their relative focus on healthcare, inasmuch as three are supermarket based (with a predominant focus on grocery provision) and others offer an extensive range of non-healthcare (eg, beauty) products. Anecdotally, these chains also differ from the perspective of employees in relation to their working environments, in particular around the relative emphasis on enforcing business targets. It would be expected, therefore, that the resultant variation in organisational culture would have some bearing on the productivity of different organisations.

The significant association identified between dispensing volume and staffing levels and skill-mix is likely to be driven by the need to meet the demands of high patient numbers while minimising waiting times. However, while demonstrating univariable associations with MUR volume, higher staffing levels and greater skill-mix were not retained in the final multivariable model for MUR volume. It might be expected that a more diverse skill-mix including pharmacy technicians and accuracy checkers would free up a pharmacist from elements of the dispensing process, giving them time for extended clinical services such as MURs. However, faced with the demands of a busy dispensary, the needs of walk-in patients are often prioritised over the provision of other services to minimise waiting times and meet patient expectations. The 400/year cap may also be influencing this association, preventing pharmacies with greater capacity to undertake MURs from doing so. Again, this highlights the need for more in-depth exploration of the mechanisms of association between these organisational factors and service volume.

Concerns over profiteering and the quality of MURs provided by large pharmacy chains have not dissipated in the 12 years since their introduction. In 2016, a national newspaper investigating the scheme reported that it was open to abuse by unscrupulous organisations pressuring pharmacists into conducting unnecessary MURs.[36] This

reflects findings from previous studies suggesting that, in order to meet organisational service targets, pharmacists sometimes offered MURs to patients without complex needs who were unlikely to benefit.[12 13] The finding from this current study that MUR volume is not (or, in some cases, inversely) related to local health need provides no evidence to counteract such perceptions of the prioritisation of quantity (MUR volume) over quality (targeting health needs) by large pharmacy corporations. It is possible that the 400/year cap on the number of MURs that are remunerated limits the extent to which this service can meet local need in some areas. However, a removal of this cap may increase existing pressure on pharmacists already suffering from work overload.[16 17 37 38]

These findings and others suggest that a fee-for-service payment model may not be the most appropriate for cognitive services, rather that payment mechanisms should incentivise quality, for example, on the basis of outcomes. Further changes to the community pharmacy contractual framework in England introduced in 2016/2017 will see, for the first time, the introduction of payments for meeting a set of defined quality criteria.[39] However, pharmacy outcome metrics are complex to define and implement,[40 41] and it remains to be seen whether these changes will impact on the balance struck within these private sector healthcare providers between the need to generate profit and the quality of services delivered. Further research is needed to develop and evaluate measures of service quality suitable for adoption by pharmacy commissioners.

**Author affiliations**
¹Centre for Biostatistics, Division of Population Health, Health Services Research and Primary Care, Faculty of Biology Medicine and Health, The University of Manchester, Manchester, UK
²Centre for Pharmacy Workforce Studies, Division of Pharmacy and Optometry, Faculty of Biology Medicine and Health, The University of Manchester, Manchester, UK
³Collaboration for Leadership in Applied Health Research and Care Greater Manchester, Centre for Primary Care, Division of Population Health, Health Services Research and Primary Care, Faculty of Biology Medicine and Health, The University of Manchester, Manchester, UK
⁴Division 5, NIHR Comprehensive Research Network – Eastern, Norwich, UK
⁵Controlled Drugs Team, NHS England (North Region) Cheshire and Merseyside, Liverpool, UK
⁶College of Pharmacy, California Northstate University, Elk Grove, California, USA

**Acknowledgements** We would like to thank all study participants and other pharmacy stakeholders who helped to facilitate this study. We would also like to acknowledge the support of staff at the NHS BSA and NHS England in relation to access to community pharmacy activity data.

**Contributors** MH contributed to study design (quantitative sampling and analysis), undertook sampling, conducted all quantitative analysis and prepared the results for publication, drafted methods and findings. EIS contributed to study design, provided ongoing advice and guidance to study and revised paper critically for important intellectual content. FB contributed to study design and revised paper critically for important intellectual content. RE contributed to the design of research instruments, conducted fieldwork and revised paper critically for important intellectual content. AW contributed to study design, obtained access to and prepared socioeconomic and demographic datasets and revised paper critically for important intellectual content. DH and KH contributed to study design, stakeholder engagement activities, provided ongoing advice and guidance to study and revised paper critically for important intellectual content. SJ, chief investigator, conceived and designed the study, contributed to data collection, analysis and interpretation, drafted the introduction and discussion and revised and finalised the paper for publication. All authors gave final approval of the version to be published and agreed to be accountable for all aspects of the work in ensuring that questions related to the accuracy or integrity of any part of the work are appropriately investigated and resolved.

**Funding** This paper presents independent research funded by the UK National Institute for Health Research (NIHR) Health Services and Delivery Research (HS&DR) programme – project number 11/1025/05.

**Disclaimer** The views and opinions expressed are those of the authors and do not necessarily reflect those of the NHS, the NIHR, NETSCC, the HS&DR programme or the Department of Health.

**Competing interests** None declared.

**Ethics approval** National Research Ethics Service (NRES) Committee West Midlands – Edgbaston.

**Provenance and peer review** Not commissioned; externally peer reviewed.

**Data sharing statement** No additional data are available.

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
