## [Reviewer comments · BMJ Open]

ARTICLE DETAILS

TITLE (PROVISIONAL)	ORGANISATIONAL AND EXTRA-ORGANISATIONAL DETERMINANTS OF VOLUME OF SERVICE DELIVERY BY ENGLISH COMMUNITY PHARMACIES: A CROSS-SECTIONAL SURVEY AND SECONDARY DATA ANALYSIS
AUTHORS	Hann, Mark; Schafheutle, Ellen; Bradley, Fay; Elvey, Rebecca; Wagner, Andrew; Halsall, Devina; Hassell, Karen; Jacobs, Sally

VERSION 1 – REVIEW

REVIEWER	Christine Bond University of Aberdeen UK
REVIEW RETURNED	06-Jun-2017

GENERAL COMMENTS	This is a very interesting well written paper. I have a few comments detailed below. the main change I would request is more information on the survey Abstract 1. No comments Introduction 2. This is comprehensive and reflects the authors inside knowledge of the topic. I thought the reference 9 (dated 2007) should be updated as I don't think GPs opinions in 2007 just after MURs were introduced is relevant now 10 years later 3. Page 5 line 17- please be explicit on the exact relationship between ownership and organisational culture. Method 4. Page 6 lines 13-23- this is slightly ambiguous, as it refers to Figure 1 of Bradley et al yet also appears to refer to Figure 1 of this paper 5. and page 6 lines 37-45 are also very unclear ie all of clause (c) 6. And why is depression not part of mental health 7. Page 6 line 50- same as comment 4 above 8. Need a bit more general information on the logistics of acquiring , storing and managing the routine data from the various sources. Was any data cleaning needed? how was it linked? how were the categories agreed for continuous variables 8.the method for the survey needs much more detail. What was the sampling frame for the community pharmacies? are the names and addresses in the public domain? what were the eligibility criteria? How were the selected pharmacies identified, approached (email, letter?) and by whom?
---

	9. How was it identified that four large chains declined participation? were head offices contacted? 10. the sample size calculation should be based on an a priori number not a back calculation base don what was available after the chains declined ? 11. How was the survey developed and by whom? I would prefer to see a little more detail in the text and not all reserved for the supplementary material. 12. Was the survey piloted?. 13. How was survey sent? What were response options? Results refer to online returns. Was there a reminder? 14. Page 7 line 40 typo 'areal' 15. Page 7 Line 52 Explain how probability weights were derived- on what basis? Results 16. Page 8 line 36 refers to 11033 pharmacies yet earlier in abstract 10500 are detailed. 17. page 9 line 7 Why weren't quintiles used for grouping the outcome 18. Page 10 line 45/46 Distance selling pharmacies needs explaining- have these been mentioned before? 19. Page 9 line 48 4/3? 20. Line 55 PSO in text needs to be in full- should not need to go to Table to understand 21. Page 12 line 48- what was basis for these groupings? 22. Page 12 line 50- style point could delete also at end of sentence 23. page 13 line 4 - suggest reword to 'are associated with' as 'are more likely to' seems to imply more causality Discussion 24. I think the limitations of the poor response rate are not fully acknowledged. It is about the bias that is introduced from non responders that needs to be discussed, rather than being about the absolute number per se. If a pilot had been undertaken or a reminder sent (unclear if this was the case) a better response might have been achieved. 25. Figure 1- could different symbols be used for the two lines eg crosses and dots so easier to differentiate in black and white? 26. Figure 2 the horizontal axis labels are unclear. Should say I assume number of MURS and the numbers are too small to read
--	--

REVIEWER	William Doucette University of Iowa United States of America
REVIEW RETURNED	07-Jun-2017

GENERAL COMMENTS	1. Overall this is a well-designed study of organizational influences on the delivery of dispensing and medication review services in the UK. the use of secondary and survey data in parallel analyses is novel. 2. The findings provide useful information about organizational and regional determinants of pharmacy services. 3. For the analyses of the survey data it is not clear that the data for the pharmacy service volume are from the same time period. That is, were the services being delivered at the same time that the survey data were collected?
---

	This should be clarified. 4. Can the authors provide explanation for how the categories were determined for the number of MURs provided? 5. it seems that the number of medications would be a big driver of MURs, as is shown in the findings. The authors treat prescriptions dispensed more as a control variable, though it could represent an important organizational variable. Greater discussion of the association between dispensing volume and MUR volume could be informative to readers. 6. Not being from the UK, it would be helpful to have a bit more discussion/description of the large chains there. Are they similar? Are some with more of a "healthcare provider" image than others? Other characteristics that may be important to these findings? 7. I think Conclusions should be about the findings of a study, and less so about other work. Since this study did not assess the quality of pharmacy services, but only quantity, I suggest that the 2nd to last sentence of the Conclusion in the abstract be revised to remove the part about quality of services. It is OK to discuss this issue, since it puts these findings into context, but not as a concluding statement.
--	---

VERSION 1 – AUTHOR RESPONSE

Reviewer: 1

This is a very interesting well written paper. I have a few comments detailed below. the main change I would request is more information on the survey

Abstract

1. No comments

Introduction

2. This is comprehensive and reflects the authors inside knowledge of the topic. I thought the reference 9 (dated 2007) should be updated as I don't think GPs opinions in 2007 just after MURs were introduced is relevant now 10 years later

Response: Despite thorough searching by members of the research team, we have been unable to locate any recent research evidence of GPs' opinions of MURs. We know from other studies of pharmacist-GP integration that collaborative working between the professions remains variable at best. We have revised the text to make the timing of the opinions stated clear and have included the only other reference we could identify in this area from research conducted in 2008/9. (p4, lines 29-30)

3. Page 5 line 17- please be explicit on the exact relationship between ownership and organisational culture.

Response: This relationship has now been made explicit. (p5, lines 11-13)

Method

4. Page 6 lines 13-23- this is slightly ambiguous, as it refers to Figure 1 of Bradley et al yet also appears to refer to Figure 1 of this paper

Response: Have clarified in text (p6, lines 13-14)

5. and page 6 lines 37-45 are also very unclear ie all of clause (c)

Response: Have simplified clause (c) (p6, lines 23-24)

6. And why is depression not part of mental health

In the QOF data, mental health indicators refer to schizophrenia, bipolar affective disorder and other psychoses and are distinct from indicators for depression.

Response: Have added text to clarify. (p6, lines 26-27)

7. Page 6 line 50- same as comment 4 above

Response: Have again clarified in text (p6, line 32 and p7, line1)

8. Need a bit more general information on the logistics of acquiring, storing and managing the routine data from the various sources. Was any data cleaning needed? how was it linked? how were the categories agreed for continuous variables

Response: Data from the PSNC and BSA website were downloaded directly. Those from the BSA were obtained through a secure portal as were the demographic, socio-economic and health-needs data. It was not possible to clean the data as there was nothing to cross-check it with. However, for the analysis of pharmacy activity in the national dataset, pharmacies were excluded with extremely high or low annual dispensing volumes (<5000 or >1,000,000 items/year) relative to other pharmacies (outliers, n=60) and those for which a full set of (linked) data for 2012/13 were unavailable. Data linkage is already described in the text (p6, lines 28-30). MUR volume categorisation is described below (response to comment 17). All data were stored securely on encrypted university network drives.

Where necessary, additions have been made to the text to clarify each of the above points. (pages 6 and 7)

8.the method for the survey needs much more detail. What was the sampling frame for the community pharmacies? are the names and addresses in the public domain? what were the eligibility criteria? How were the selected pharmacies identified, approached (email, letter?) and by whom?

Response: More detail has now been provided about the sampling frame, how contact details were obtained, inclusion/exclusion criteria and method of contact. (p7, lines16-24)

9. How was it identified that four large chains declined participation? were head offices contacted?

Response: Additional detail added to text to describe contact with head offices and subsequent exclusion of four chains from sampling frame. (p7, lines 19-24)

10. the sample size calculation should be based on an a priori number not a back calculation based on what was available after the chains declined ?

Response: This is not a back calculation. Originally, we had determined that we could detect a correlation as small as 0.12, accounting for non-response, in the 5 primary care areas we originally had agreement from to supply pharmacy contact details. When it became apparent that the original design was not going to work (given the declining chains), we expanded the number of research sites from 5 to 9 and re-calculated the power/ detectable correlation based on the known number of eligible pharmacies in the, now, 9 primary care areas.

This was a particularly complex study to conduct in that engagement with community pharmacies was problematic and there were difficulties obtaining permission to access the pharmacy activity data. As a result we had to make changes to the original protocol as the study progressed. For reasons of readability and paper length, and in order not to over-complicate things, we have tried to report only what we finally did rather than explain what we started off doing and the changes that were made on the way. However we recognise that this may raise questions such as that posed by the reviewer. A full explanation of the methods will be available on publication of the final report.

11. How was the survey developed and by whom? I would prefer to see a little more detail in the text and not all reserved for the supplementary material.

Response: The detail of the information collected by the survey is provided in Table 2 and not supplementary material. This was an error on my part and has now been corrected. Additional information about how items were developed has been provided. (p8, lines 1-6)

12. Was the survey piloted?.

Response: Additional information has been added about piloting. (p8, lines6-8)

13. How was survey sent? What were response options? Results refer to online returns. Was there a reminder?

Response: Information has been added regarding survey distribution, returns and reminders. (p7, lines 27-30)

14. Page 7 line 40 typo 'areal'

Response: This is not a typo. Areal is the adjective form of area and means pertaining to the area of something.

15. Page 7 Line 52 Explain how probability weights were derived- on what basis?

Response: Explanation of calculation of probability weights now provided. (p8, lines 23-25)

Results

16. Page 8 line 36 refers to 11033 pharmacies yet earlier in abstract 10500 are detailed.

Response: 11,033 was the number of pharmacies with dispensing/ MUR data for the financial year 4/2012 to 3/2013. This number was used to generate Figure 2 in the paper. 10,454 (the ≈10,500 in the Abstract) was the final number of pharmacies used in the national regression models with outliers removed (with the restriction that 5,000 < number of dispensed items < 1,000,000) and for which a full set of (linked) data was available. Clarification of this has now been provided throughout the text.

17. page 9 line 7 Why weren't quintiles used for grouping the outcome

Response: We believe that there is no greater justification for using quintiles than the categorisation we have used. We decided to use objective 'MURs per month' criteria (equivalent to <1 per month up to >1 per day). Given the unusual distribution, using quintiles might easily have left us with 0-200, 200-395, 395-399, 400, 401+ which is no less arbitrary. The rationale behind the choice of categorisation has now been provided in the text. (p10, line 1)

18. Page 10 line 45/46 Distance selling pharmacies needs explaining- have these been mentioned before?

Response: Definition now provided (p11, line11)

19. Page 9 line 48 4/3?

4/3 is the exact fraction (equivalent to 1.33 recurring).

Response: We have changed this in the text to 1.3 to avoid further confusion. (p11, line 12)

20. Line 55 PSO in text needs to be in full- should not need to go to Table to understand

Response: PSO has now been defined earlier in text as well as in table – is this sufficient? (p8, line 4)

21. Page 12 line 48- what was basis for these groupings?

Response: These groupings have already been defined earlier in the text (original p9). And further explanation given (see response to reviewer's comment 17 above).

22. Page 12 line 50- style point could delete also at end of sentence

Response: Have deleted 'also' (p13, line 24)

23. page 13 line 4 - suggest reword to 'are associated with' as 'are more likely to' seems to imply more causality

Response: Have added word 'also' instead of suggested change to avoid implication of causality whilst still reading clearly. (p14, line 2)

Discussion

24. I think the limitations of the poor response rate are not fully acknowledged. It is about the bias that is introduced from non responders that needs to be discussed, rather than being about the absolute number per se. If a pilot had been undertaken or a reminder sent (unclear if this was the case) a better response might have been achieved.

Response: The possibility of non-response bias has already been acknowledged but this has now been further discussed including information on how we tried to mitigate against this. (p16, lines 5-9)

25. Figure 1- could different symbols be used for the two lines eg crosses and dots so easier to differentiate in black and white?

Response: We have now used different symbols for the tow lines and uploaded a revised version of this figure.

26. Figure 2 the horizontal axis labels are unclear. Should say I assume number of MURS and the numbers are too small to read

Response: We have now included a label for the x-axis and increased the font size for the numbers and uploaded a revised version of this figure.

Reviewer: 2

Overall this is a well-designed study of organizational influences on the delivery of dispensing and medication review services in the UK. the use of secondary and survey data in parallel analyses is novel.

2. The findings provide useful information about organizational and regional determinants of pharmacy services.

3. For the analyses of the survey data it is not clear that the data for the pharmacy service volume are from the same time period. That is, were the services being delivered at the same time that the survey data were collected? This should be clarified.

Response: This has now been clarified in the text. (p8, lines 10-11)

4. Can the authors provide explanation for how the categories were determined for the number of MURs provided?

Response: As above – see response to reviewer 1, comment 17.

5. it seems that the number of medications would be a big driver of MURs, as is shown in the findings. The authors treat prescriptions dispensed more as a control variable, though it could represent an important organizational variable. Greater discussion of the association between dispensing volume and MUR volume could be informative to readers.

Response: In the discussion we already discuss the association between dispensing volume and MUR volume and why that occurs. We also discuss that this association may be complex: “The volume of MURs conducted was strongly associated with dispensing volumes overall, suggesting that in high dispensing volume pharmacies with a higher patient footfall (and more staff and more diverse skill-mix) there are increased opportunities to offer MURs. However, comparably high numbers of MURs conducted in the low dispensing volume supermarket pharmacies suggests other influences may be at play. Previous qualitative research has suggested that pressures on pharmacists to meet targets for MURs are more stringent in these larger organisations [11, 12] and this may be a contributing factor.”

It is not clear what additional discussion may be required by the reviewer.

6. Not being from the UK, it would be helpful to have a bit more discussion/description of the large chains there. Are they similar? Greater discussion of the association between dispensing volume and MUR volume could be informative to readers.

Response: Further description of the large chains in the UK has now been provided in the discussion with consideration of how their differing cultures might be expected to influence the findings. (p17, lines 16-21)

7. I think Conclusions should be about the findings of a study, and less so about other work. Since this study did not assess the quality of pharmacy services, but only quantity, I suggest that the 2nd to last sentence of the Conclusion in the abstract be revised to remove the part about quality of services. It is OK to discuss this issue, since it puts these findings into context, but not as a concluding statement.

Response: Rather than remove the part of this sentence about quality, we have clarified that we are talking specifically here about the appropriate targeting of services which derives directly from the findings of this study. In addition, we have added clarity around this in the discussion. (p2, line 29 and p3, line 1)

VERSION 2 – REVIEW

REVIEWER	Christine Bond University of Aberdeen, U.K.
REVIEW RETURNED	09-Aug-2017

GENERAL COMMENTS	Page 6 line 9 It is still not clear that this data is available in the public domain for anyone to download. It is important to clarify this for international readers  - Page 9 line 7 small pint but personally I prefer the standard approach of reporting results without any additional interpretation, so in that basis use 'Unsurprisingly' at the start of this line is inappropriate. - Page 10 line 1 I know the authors have justified this approach in their response but it still seems to have no basis and I think there should at least be some comment in this in the Discussion. - Page 10 lines 16/17 starting 'the percentage ' though to the colon on line 17 could be belated as they are repeated immediately but with the level of detail I was looking for when I read the first sentence - Page 11 line 19/page 12 lines 1-2 should be in Discussion not results - Page 16 line 9/10 the fact that the study has power because of the numbers returned is a completely different point from the non response rate which potentially leads to unknown biases. As written, the text implies that the non response did not matter because the study still had power. - Page 16 lines 17 these finding feel slightly counter intuitive and I wonder if the authors could speculate on any possible reasons for this finding. Lines 31-33 suggest volume of MUR is linked to Rx volume, which is linked to need. - Page 17 Line 18 don't understand the sense of line 18 in the newly added text- needs rewording for clarity. - Page 17 line 22 starts off referring to 'Analysis of the survey data' somehow implying nine of the survey data has yet been discussed but I don't think this is true as all organisational culture information comes from the survey? - Page 17 lines 30-32 speculate on reasons why there might be financial incentives to prioritise dispensing over MURs. Again for an international readership could the authors estimate how many prescriptions would need to be dispensed to equate to a fee similar to that of say 100 MURs? - Page 18 line 9 suggest text in brackets better as 'or is even inversely' or the text is more specific. Was it or was it not inversely related?
---

VERSION 2 – AUTHOR RESPONSE

Reviewer(s)' Comments to Author:

Page 6 line 9 It is still not clear that this data is available in the public domain for anyone to download. It is important to clarify this for international readers

Response: This has now been clarified in the text. [p6, line 12]

Comment: Page 9 line 7 small pint but personallly I prefer the standard approach of reporting results without any additional interpretation, so in that basis use 'Unsurprisingly' at the start of this line is inappropriate.

Response: The word 'unsurprisingly' has now been deleted. [p9, line7]

Page 10 line 1 I know the authors have justified this approach in their response but it still seems to have no basis and I think there should at least be some comment in this in the Discussion.

Response: I have discussed this matter in depth with the statistician who worked on the study, Mark Hann. He has advised me that there is no benefit to using quintiles above the approach we have taken. Given the unusual distribution, particularly the pronounced peak at 400, there was a risk that some of the quintile groupings would be closely clustered around 400 and would therefore not be substantially different from each other. On close consideration of the distribution, we therefore made the pragmatic decision to categorize this variable in a more meaningful way, ensuring adequate numbers in each category and incorporating a temporal element i.e. 0-12 (up to 1/month), 13-200, 201-365, >365 (more than 1 per day). We accept that it is possible that using quintiles may have produced a different result. We have therefore now expanded the justification of the method used in the methods section (p 9 line 28 to p10 line 2) and have commented on this in the limitations section of the discussion (p 16, lines 11-13). I hope that this is now acceptable.

Comment: Page 10 lines 16/17 starting 'the percentage ' though to the colon on line 17 could be belated as they are repeated immediately but with the level of detail I was looking for when I read the first sentence

Response: This sentence has now been revised, deleting much of the content before the colon but retaining the detail that these were the strongest associations with dispensing volume. [p10, lines 17/18]

Comment: Page 11 line 19/page 12 lines 1-2 should be in Discussion not results

Response: This has now been moved to the discussion [p17, lines 21-25]

Comment: Page 16 line 9/10 the fact that the study has power because of the numbers returned is a completely different point from the non response rate which potentially leads to unknown biases. As written, the text implies that the non response did not matter because the study still had power.

Response: Thank you for pointing this out – it had not been our intention to imply this. The text has now been revised to clarify this. [p16, lines 9/10]

Comment: Page 16 lines 17 these finding feel slightly counter intuitive and I wonder if the authors could speculate on any possible reasons for this finding. Lines 31-33 suggest volume of MUR is linked to Rx volume, which is linked to need.

Response: We agree that these findings are counter-intuitive. We have included some new text to speculate on the reasons for this finding although the mechanism is far from clear and may be complex. [p16, lines 25-29]

Comment: Page 17 Line 18 don't understand the sense of line 18 in the newly added text- needs rewording for clarity.

Response: The newly added text has been revised which will hopefully add clarity. [p17, lines 28-33]

Comment: Page 17 line 22 starts off referring to 'Analysis of the survey data' somehow implying nine of the survey data has yet been discussed but I don't think this is true as all organisational culture information comes from the survey?

Response: By deleting the first sentence of this paragraph (p18, lines 3-4), this implication has been removed without losing any meaning from the rest of the paragraph.

Comment: Page 17 lines 30-32 speculate on reasons why there might be financial incentives to prioritise dispensing over MURs. Again for an international readership could the authors estimate how many prescriptions would need to be dispensed to equate to a fee similar to that of say 100 MURs?

Response: This part of the discussion was not intended to imply that there were financial incentives to prioritise dispensing, rather that dispensing was prioritised to meet the demands of walk-in patients. This text has now been revised to clarify this. [p18, lines 11-14]

Comment: Page 18 line 9 suggest text in brackets better as 'or is even inversely' or the text is more specific. Was it or was it not inversely related?

Response: The wording in brackets has been revised for clarification. [p18, line 24]